# Re-Examining the Role of TNF in MS Pathogenesis and Therapy

**DOI:** 10.3390/cells9102290

**Published:** 2020-10-14

**Authors:** Diego Fresegna, Silvia Bullitta, Alessandra Musella, Francesca Romana Rizzo, Francesca De Vito, Livia Guadalupi, Silvia Caioli, Sara Balletta, Krizia Sanna, Ettore Dolcetti, Valentina Vanni, Antonio Bruno, Fabio Buttari, Mario Stampanoni Bassi, Georgia Mandolesi, Diego Centonze, Antonietta Gentile

**Affiliations:** 1Synaptic Immunopathology Lab, IRCCS San Raffaele Pisana, 00166 Rome, Italy; diego.fresegna@gmail.com (D.F.); Silvia.Bullitta@uniroma2.it (S.B.); msllsn00@uniroma2.it (A.M.); livia.guadalupi@gmail.com (L.G.); valentina_vanni@hotmail.it (V.V.); georgia.mandolesi@uniroma5.it (G.M.); antonellag79@gmail.com (A.G.); 2Synaptic Immunopathology Lab, Department of Systems Medicine, Tor Vergata University, 00133 Rome, Italy; f.rizzo@med.uniroma2.it (F.R.R.); balletta.sara@gmail.com (S.B.); krizia.sanna@live.it (K.S.); ettoredolcetti@hotmail.it (E.D.); antonio.bruno91@yahoo.it (A.B.); 3Department of Human Sciences and Quality of Life Promotion, University of Rome San Raffaele, 00166 Roma, Italy; 4Unit of Neurology, IRCCS Neuromed, Pozzilli (Is), 86077 Pozzilli, Italy; f.devito.molbio@gmail.com (F.D.V.); silviacaioli@yahoo.it (S.C.); fabio.buttari@gmail.com (F.B.); m.stampanonibassi@gmail.com (M.S.B.)

**Keywords:** TNFR1, TNFR2, TNF therapy, demyelination, neurodegeneration, synaptic damage, experimental autoimmune encephalomyelitis, cuprizone, Theiler’s murine encephalomyelitis virus

## Abstract

Multiple sclerosis (MS) is a common neurological disorder of putative autoimmune origin. Clinical and experimental studies delineate abnormal expression of specific cytokines over the course of the disease. One major cytokine that has been shown to play a pivotal role in MS is tumor necrosis factor (TNF). TNF is a pleiotropic cytokine regulating many physiological and pathological functions of both the immune system and the central nervous system (CNS). Convincing evidence from studies in human and experimental MS have demonstrated the involvement of TNF in various pathological hallmarks of MS, including immune dysregulation, demyelination, synaptopathy and neuroinflammation. However, due to the complexity of TNF signaling, which includes two-ligands (soluble and transmembrane TNF) and two receptors, namely TNF receptor type-1 (TNFR1) and type-2 (TNFR2), and due to its cell- and context-differential expression, targeting the TNF system in MS is an ongoing challenge. This review summarizes the evidence on the pathophysiological role of TNF in MS and in different MS animal models, with a special focus on pharmacological treatment aimed at controlling the dysregulated TNF signaling in this neurological disorder.

## 1. Introduction

Multiple sclerosis (MS) is a chronic neurodegenerative disease of the central nervous system (CNS) associated with uncontrolled inflammation and autoimmunity, as a result of an attack against central myelin by blood-borne autoreactive T lymphocytes [1]. The natural history of the disease is very heterogeneous. In many cases (about 87%), the clinical course of MS is unpredictable and characterized by repeated episodes of neurological deficits in the early stages (relapsing–remitting MS—RRMS) followed by a progressive neurological decline after 15–20 years since the diagnosis (secondary progressive MS—SPMS). Approximately 10–15% of MS patients show a progressive course from the onset, with rare or nonexistent clinical relapses (primary progressive MS—PPMS) [2]. Despite the clinically different phenotypes, which likely underlie distinct pathological processes [3,4], MS brains share some neuropathological hallmarks, which include multiple inflammatory demyelinated plaques distributed throughout the neuroaxis and characterized by infiltrating T cells, activated microglia and astroglia, synaptic loss and neurodegeneration [5,6]. The etiology of MS remains unclear; however, it can be considered a multifactorial disease that includes a genetic predisposition combined with environmental influences. A complex and altered peripheral interplay between cells of the innate and the adaptive immune system, is currently regarded as the *primum movens* in MS pathogenesis [7]. Autoreactive T cells, which have escaped the mechanisms of immune tolerance orchestrated by T regulatory cells (Tregs), are supposed to be stimulated by antigen-presenting cells (APCs) with putative exogenous antigen and equipped with molecular mimicry for epitopes of basic myelin protein expressed by CNS oligodendrocytes [7]. The concomitant breakdown of the blood–brain barrier (BBB) caused by the mounting inflammatory reaction facilitates the migration of such activated and autoreactive lymphocytes into the CNS, where they trigger and then fuel the subsequent myelin and neuronal damage. The release and the diffusion of a number of cytokines such as tumor necrosis factor (TNF), activate resident microglia with the consequent additional release of proinflammatory cytokines, thus further priming infiltrated lymphocytes [5]. Of note, CNS inflammation is proposed to drive neuronal damage in MS, by altering synaptic transmission from the very early stages of the disease [8].

TNF is a pleiotropic cytokine increasingly recognized to regulate important physiological processes not only in the immune system, but also in the brain [9,10,11]. Evidence for TNF involvement in MS includes the identification of TNF in astrocytes, microglia, and endothelial cells, preferentially in acute and chronic active MS brain lesions and in the cerebrospinal fluid (CSF) of MS subjects [12,13,14]. TNF has a complex, and presumably multifaceted, role in the immunopathogenesis of MS, as suggested by both animal experiments and clinical studies in humans, including the failure of an anti-TNF clinical trial [15,16,17].

Here, we review the pleiotropic functions of TNF in both the immune system and the CNS, with a focus on those processes that are dysregulated in MS, such as Treg function, neuronal transmission, BBB permeability and myelination. Moreover, we provide an updated overview of data from different animal models of MS and from MS patients about the role of TNF and its receptors in MS pathology, proposing a re-examination of therapies targeting TNF signaling in MS.

## 2. TNF Cellular Expression and Signaling

TNF is the prototypical cytokine of the so-called TNF superfamily, which includes 19 ligands, while the TNF receptor (TNFR) superfamily (TNFRSF) consists of 29 related receptors [18]. Newly synthesized TNF is expressed initially as a transmembrane protein (tmTNF), which requires proteolytic cleavage by the TNF converting enzyme (TACE, also named ADAM17) to release soluble TNF (solTNF, monomeric 17 kDa) [19]. Both tmTNF and solTNF are biologically active and their synthesis and signaling depend on several factors, such as cell type and extent of the stimulus inducing TNF production [20].

TNF family members exert their cellular effect through two distinct surface receptors, the TNF Receptor Superfamily Member 1A (TNFRSF1A-TNFR1), also known as p55 or p60, and TNF Receptor Superfamily Member 1B (TNFRSF1B-TNFR2), also known as p75 or p80 [21]. Throughout the text we will refer to these as TNFR1 and TNFR2. TNFR1 and TNFR2 differ in their expression profiles, ligand affinity, cytoplasmic tail structure, and downstream signaling pathway activation [22]. Only TNFR1 contains a cytoplasmic death domain and may directly induce apoptosis. TNFR1 is constitutively expressed in most tissues, whereas expression of TNFR2 is highly regulated and is restricted to specific cell types, such as neurons, microglia, oligodendrocytes, T cells and endothelial cells [9,23]. TNFR1 primarily promotes inflammation and tissue degeneration, whereas TNFR2 mediates local homeostatic effects, such as cell survival and tissue regeneration [10]. TNFR1 shows high affinity towards both forms of TNF, with a preference for solTNF, while TNFR2 is preferentially engaged by tmTNF [24]. The membrane-bound forms of both receptors are also substrate for the proteolytic cleavage driven by TACE, with the production of soluble TNF receptor (solTNFR) [18]. This process is an important self-regulatory mechanism to prevent exaggerated damage and may contribute to the regulation of cellular responsiveness to TNF. Indeed, solTNFR acts as an intrinsic scavenger of circulating TNF, thereby reducing its availability and its increased rate of production limits TNF-signaling through surface receptors.

The TNF-TNFR signaling pathways are complex and wide-ranging in different cell types and precise circumstances, thereby accounting for TNF pleiotropic nature of action [25]. Despite their divergent functions, both TNFR1 and TNFR2 activate the transcription factor nuclear factor-κB (NF-κB). Of note, NF-κB is a major regulatory transcription factor with a pivotal role in inducing genes involved in inflammation [26]. Indeed, NF-κB nuclear translocation activates transcription of many genes for proinflammatory cytokines, like IL-1β, IL-6 and TNF itself [27,28]. Thus, TNF both activate and is activated by NFκB, creating a positive regulatory loop that amplifies and perpetuates local inflammation [29].

## 3. Pathophysiological Role of TNF in the Immune System and the Brain

TNF has long been regarded as the key regulator of the inflammatory response, as shown in sepsis [30]. Nowadays, the spectrum of TNF functions has increasingly broadened to include its involvement in several biological functions other than the immune surveillance, like homeostatic immune activity [31]. More strikingly, the recognition that TNF and its receptors are constitutively expressed to a variable extent by brain-resident cells, has paved the way to address its involvement in normal and diseased brain functioning. In this respect, the use of transgenic mice, though providing some contradictory results, has been instrumental to improve our knowledge about the different functions of TNF in pathophysiological conditions. Studies in TNF-deficient mice have shown that this cytokine is required for the proper lymphoid organ organization and for host defense responses against pathogens [32,33]. TNF-deficient mice are viable and show neither major phenotypic alterations nor massive brain morphological changes, apart from some abnormalities described in the developing hippocampus [34].

Below, we revise main findings regarding the involvement of TNF in both immune system and brain functioning.

### 3.1. TNF Role in the Immune System

Cytokines provide essential communication signals to the highly motile cells of the immune system. Upon cellular activation, TNF is mostly produced by cells of the innate immune system like mononuclear phagocytes but it is also expressed by neutrophils and lymphocytes [35,36]. This wide cellular expression in the immune system makes TNF a crucial regulator of autoimmune responses, which explains the development of a plethora of anti-TNF drugs for autoimmune disorders (see later paragraph 4). Due to the complexity of the immune responses mediated by TNF and for the aim of this review, we provide a brief overview on TNF role in T cell-subset function and we refer the reader to more detailed reviews on this topic [37,38].

TNF regulates many aspects of T cell biology, like proliferation, priming and survival and/or apoptotic fate [39]. A relevant aspect is that most of T cell functions regulated by TNF are largely dependent on TNFR2. Indeed, while TNFR1 is almost ubiquitously expressed in all the cells of the immune system, TNFR2 seems to be selectively expressed by T lymphocytes, in particular Tregs [40,41]. TNF induces the proliferation of thymocytes [42] (and acts as a mitogen for unstimulated cells [43] or activated T cells [44]), by acting as a costimulatory molecule of TCR, through activation of TNFR2 signaling [45,46]. In addition, TNF has been identified as a crucial factor in the recruitment and activation of antigen-presenting cells (APCs), thereby enhancing T cell activation in situ [47].

TNF is known to play an important role in the conclusion of lymphocyte responses, as indicated by the ability of this molecule to promote activation-induced cell death in both CD4+ and CD8+ T cells, mainly through TNFR1 [48]. However, under specific inflammatory conditions, TNFR2 can promote or support T cell apoptosis [37,49], suggesting an involvement in lymphocyte clonal contraction.

It has recently become apparent that TNF can exert a strong activity on CD4+ Tregs, which are a subset of CD4+ T cells that help to prevent or treat autoimmunity by maintaining self-tolerance, immune homeostasis, and suppression of cytotoxic T cells [50]. Mounting evidence indicate a crucial role of TNFR2 in Treg biology, supporting the development of drugs specifically targeting TNFR2 to potentiate or weaken Treg functions in autoimmune disorders [38]. Indeed, TNFR2 is upregulated in activated Tregs [41] and can promote itself the proliferation and expansion of Tregs [51,52]. By contrast, TNFR1 is barely detectable in these cells [53,54]. It has been shown that human Tregs, as well as thymic and peripheral murine CD4+CD25+ Tregs, express remarkably high levels of TNFR2 relative to CD4+CD25− effector T cells [41]. TNF increases proliferation, survival, stability, expression of CD25, Foxp3, and activation markers, as well as suppressive function of mouse Tregs [55,56,57]. Many of these effects of TNF, in particular on proliferation, have been reproduced with human Tregs [58,59]. In TNFR2 KO mice, although the number and function of Tregs were comparable with wild-type mice, Tregs failed to expand when stimulated under inflammatory conditions either in vivo or in vitro [55,60]. This suggests that, under noninflammatory conditions, TNF is not required for thymic Tregs to maintain immune homeostasis [55]. Moreover, an elegant study using a double TNF/TNFR2 KO engineered to conditionally ablate TNFR2 in T cells showed that Treg suppressive functions in vitro were significantly impaired [61].

While most of the studies on human and murine cells have addressed the effects of TNF on T cells, a poorly explored and clarified issue regards the role of T cell-derived TNF in pathophysiological conditions [39]. T cells synthesize and release both tmTNF and solTNF, and this translates into the activation of both protective and harmful pathways, which can vary according to the specific context [62]. Despite some inconsistencies and possible synergistic effects of the two receptors as elsewhere reviewed [39], TNFR2 appears to play a crucial role in T cell pathological functions.

### 3.2. TNF Role in the Brain

TNF is constitutively expressed at low levels in the normal adult brain [63] and its expression could be influenced by the presence or absence of circulating cytokines that can cross the intact BBB in small quantities [64]. TNFRs in the brain are expressed by neurons and glia [65]. The differential patterns of localization of TNFRs in neuronal and glial cells, their state of activation and the downstream effectors, are thought to play an important role in determining whether TNF will exert beneficial or harmful effects on CNS. Under healthy status, TNF has regulatory functions on vital physiological CNS processes such as homeostatic synaptic plasticity, astrocyte-mediated synaptic transmission, and neurogenesis, which regulate, among others, learning and memory functions [66].

Glial cells, astrocytes and microglia, are the main source of TNF in the brain in both physiological and pathological conditions [67,68]. Noteworthy, astrocytes and microglia are now recognized as integral components of CNS synapses, playing critical roles for sensing and removing glutamate from the synaptic cleft, thereby limiting the duration of synaptic excitation [69]. Hence, TNF is widely recognized to be a physiological gliotransmitter involved in the communication between neurons and glial cells and, thus, in synapse regulation at multiple levels. In fact, TNF has been shown to be important for homeostatic synaptic scaling, a form of synaptic plasticity that allows for adjustment of the strength of all synapses on a neuron in response to prolonged changes in cellular activity [70,71]. Glial TNF is constitutively required for the maintenance of normal surface expression of AMPA receptors. In vitro experiments of stimulation of hippocampal pyramidal cells with increasing doses of TNF demonstrated the new insertion of AMPARs in membrane and the strengthening of glutamatergic synaptic activity [70]. Accordingly, treatment of hippocampal neurons with solTNFR, which functions as a TNF antagonist, abrogated both AMPAR membrane insertion and synaptic transmission potentiation [70]. In addition, a prolonged activity blockade (48 h) induced synaptic scaling by increasing glial release of TNF, which then acts on neurons to enhance AMPAR insertion [71]. In contrast, other studies provided evidence that scaling is a gradual and cumulative process evident after as little as 4–6 h of activity blockade [72,73].

TNF has been reported to control glutamate transmission also by modulating NMDA currents. The use of TNF KO mice has revealed the permissive role of constitutive TNF in the control of glutamate release from astrocytes in response to purinergic signaling, with the consequent enhancement of the excitatory synapses in the dentate gyrus of the hippocampus through the activation of presynaptic NMDAR [74]. Application of TNF has been reported to either increase [75,76] or decrease [77] NMDA currents on spinal cord or hippocampal neurons, respectively [78].

TNF can potentiate glutamate excitotoxicity even in conditions of subthreshold glutamate levels, directly through activation of glutamate-NMDA receptors [79] and localization of AMPA receptors to synapses [80,81], and indirectly by inhibiting glial glutamate transporters on astrocytes [82]. Interestingly, Marchetti et al., 2004 proved that cortical neurons from TNFR1 KO mice prestimulated with TNF or agonistic TNFR2-specific antibodies, were protected from glutamate-induced toxicity. Furthermore, neurons derived from TNFR2-deficient mice were susceptible to both TNF and glutamate-induced death [83]. These results suggest that TNFR2 signaling may counteract excitotoxicity, which is a hallmark of several neurodegenerative diseases, such as Alzheimer’s disease (AD) and Parkinson’s disease (PD) that, like MS, are associated with increased TNF levels in lesioned brain areas [84,85,86,87,88,89].

In addition to excitotoxicity, in the injured CNS the increased concentration of TNF released by astroglia and microglia boost neuroinflammatory responses [90]. As elsewhere reviewed, in the context of neuroinflammation different studies have focused on TNF effect on BBB permeability [91,92], showing that both peripheral and glial-derived TNF can initiate and support neuroinflammation as well as induce damage to endothelial cells [91,92,93]. In addition, TNF signaling has been largely associated to oligodendrocyte apoptosis [94] with a pivotal role played by TNFR1 [95]. The toxic role of TNF on oligodendrocyte lineage is also highlighted by the evidence of spontaneous demyelination in transgenic mice overexpressing TNF [96]. However, in pathological states microglia-derived TNF has been reported to exert homeostatic effects in tissue regeneration, favoring neuronal remyelination, as shown in a recently established toxic model of demyelination in zebrafish [97], suggesting that TNF can exert a dual role also in pathological states. Interestingly, the development of a transgenic mouse line with the selective ablation of TNF2 in oligodendrocytes has shown that signaling through TNFR2 is not necessary for physiological development of oligodendrocytes but it promotes remyelination in pathological conditions [98].

Overall data from murine studies of brain functioning implicate TNF in several CNS activities and suggest that, under specific conditions of elevated TNF levels, TNFR1 is more likely implicated in glutamatergic transmission potentiation and demyelination, while TNFR2 signaling may be involved in reparative processes.

## 4. Evidence for TNF Involvement in Pathological Hallmarks of Experimental and Human MS

Data reported in the previous chapter highlight the crucial involvement of TNF in several immune and CNS functions that are particularly relevant for MS pathogenesis. However, the centrality of TNF in the above biological functions is a double-edged sword, because, on the one hand, it clearly links TNF to MS pathology, but, on the other hand, it makes hard to dissect the contribution of TNF to each pathological hallmark of MS, namely immune dysregulation, demyelination, neuroinflammation and neurodegeneration. In the following paragraph we revise the literature addressing the role of TNF in animal models of MS and in human studies.

### 4.1. TNF in Experimental Autoimmune Encephalomyelitis (EAE)

Due to the complexity of disease manifestations and the underpinning mechanisms, no single animal model can fully recapitulate the broad spectrum of MS symptoms and pathological hallmarks. Experimental autoimmune encephalomyelitis (EAE) is the most commonly used animal model in MS research to study the autoimmune component of MS pathology [99]. Though in some cases beneficial effects of drugs tested in EAE have not been reproduced in MS, the EAE model has cogently contributed to the development of a number of first-line treatments that target the inflammatory phase of the disease [100].

EAE disease course is significantly influenced by several factors, including species, strain and the autoantigen used. In this model, autoreactive T cells are actively induced by peripheral immunization with a myelin antigen. Such T cells collect in the spleen before migrating to the CNS, where they recognize their cognate antigen on local APCs and activates, thus starting an inflammatory cascade leading to tissue injury [101,102].

In EAE, TNF has been shown to be expressed by local microglia and infiltrating macrophages [101] and by encephalitogenic T cells [103]. TNF is produced and upregulated at various stages of EAE course and in parallel with disease progression [104]. Systemic administration of TNF increases the severity of EAE, prolongs its duration and induces relapses [105]. Peripherally produced TNF can enter the circulation and cross the BBB through active transport [106] or passively after pathological BBB disruption [107]. In the EAE model, lymphocytes have been demonstrated to initiate BBB disruption by leading to alterations in the tight junction architecture and ensuing CNS vascular permeability [108]. TNF-dependent T cell activation directly influences meningeal mast cells that promote the BBB break and, together with T cells, lead to further inflammatory cell influx, myelin damage and disease severity [109,110].

The induction of EAE in transgenic mice for TNF or TNFRs has significantly contributed to understanding TNF role in MS pathophysiology (Table 1). Several studies reported that EAE induction promotes a delayed onset in TNF KO mice compared with WT mice [111,112,113], while data on disease severity have yielded mixed results. It has been demonstrated that EAE clinical deficits in TNF KO mice were equally or even more severe with extensive demyelination and higher mortality compared with WT mice [111,114]. In other reports, TNF deficient mice showed a milder EAE disease course compared to WT probably due to reduced leukocyte trafficking into the CNS in the early stages of disease development, indicating that TNF is required for initial recruitment of inflammatory cells to the CNS [112,113,115]. Of note, TNF KO mice displayed a more severe disease manifestation together with an increase in numbers of immune cell infiltration in CNS during the symptomatic phase of EAE [115]. In the same study, the role of TNF in EAE pathology was investigated by concomitant or single ablation of the cytokine in both T cells and myeloid cells. The data showed that TNF produced by T cells exacerbated EAE, while, when it was produced by myeloid cells, it accelerated EAE onset facilitating tissue damage in CNS [115], suggesting a critical and differential role of TNF in specific immune cell populations.

TNF KO mice were also used to investigate TNF involvement in BBB leakage in the EAE model. BBB permeability changes were significantly reduced in TNF KO mice with EAE compared with WT congenic animals, suggesting that TNF plays a role in promoting BBB damage induced by EAE, possibly through its known effect on the induction of adhesion molecules and chemokines [105,116]. Moreover, the extent of BBB permeability in TNF KO mice with EAE correlated with accumulation of immune cell markers in the spinal cord tissues and the clinical score [116].

Evidence of TNF pathogenic role was further provided by anti-TNF treatment. The administration of a recombinant TNFRp55 protein constructed by fusing TNFRp55 extracellular domain cDNA to a human IgG (TNFRp55-IgG1), showed a beneficial effect in both preventive and therapeutic schedule [117]. Intraperitoneal injection of TNFR-IgG prior to onset of disease signs completely prevented the neurological deficit and markedly reduced its severity. A similar effect was observed when the antagonist was delivered systemically for few days after the manifestation of the first clinical symptoms, indicating the role of TNF as a pathogenic mediator controlling the terminal effector phase. Contrasting results have been obtained with etanercept, a recombinant fusion protein comprised of the extracellular part of the human TNFR2 coupled to a human IgG1 Fc. In one study a protective effect on clinical score mediated by etanercept was observed starting the treatment three days after EAE induction [118]. In contrast, it has been reported that preventive or therapeutic treatments with etanercept, were not able to improve EAE clinical symptoms [119,120].

To better understand the role of TNF in MS pathology, several studies addressed whether individual TNFRs were involved in different processes of disease progression. Findings in transgenic mice indicate that TNFR1 plays a detrimental role in MS whereas TNFR2 has a protective one. In particular, TNFR1/TNFR2 double-KO and TNFR1 KO mice were completely protected against EAE showing a reduction in maximal disease severity compared to WT mice, whereas TNFR2 KO mice showed exacerbated disease symptoms, demyelination, enhanced Th1 cytokine production, and enhanced CD4+ T cell infiltration in the CNS [112,121,122,123]. In line with this, mice capable of making transmembrane TNF, which acts preferentially through TNFR2, but not solTNF, were protected from developing MS-like disease in the murine model of EAE, confirming that TNFR2 may exert a beneficial effect in EAE [124].

The TNFR2-mediated protective action was also supported by evidence in conditional KO mice that demonstrated the crucial role of TNFR2 in oligodendrocyte differentiation [98] and suppression of activated lymphocytes [122]. In particular, the specific ablation of TNFR2 in oligodendrocytes induced an exacerbated disease course, characterized by impairment of remyelination and significant loss of oligodendrocytes and their precursors. Importantly, treatment of the EAE mutant mice with the solTNF inhibitor XPro1595 was ineffective in the recovery of the symptoms, indicating that tmTNF can exert its beneficial effects of oligodendrocytes only in the presence of TNFR2 [98]. Moreover, lack of TNFR2 in oligodendrocytes induces an early activation of microglia, an increased BBB permeability and infiltration of immune cells in the spinal cord, leading to an accelerated onset and a higher disability peak, and to an overall increased severity of EAE [125]. This suggests that TNFR2 signaling in oligodendrocytes exert an immune-inflammatory function in EAE. In another study, TNFR2 role in myeloid cells has been addressed. TNFR2 ablation in microglia resulted in an early onset of EAE together with spinal cord inflammation and demyelination [126]. In contrast, TNFR2 ablation in monocytes/macrophages resulted in impaired peripheral immunity, reduced CNS T cell infiltration, demyelination and alleviated EAE motor disease development [126]. These results indicated that the opposite functions of microglial versus monocyte/macrophagic TNFR2 may depend to the different compartments the cells originate from and highlight the divergent role of TNFR2 in peripheral and central innate immune system.

Recent studies indicated an important role for TNF and TNFR2 in the activation of Treg cells. It was shown that Treg-TNFR2-deficient mice developed exacerbated EAE disease, indicating that intrinsic TNFR2 signaling in Tregs provides protection in CNS autoimmunity [61]. However, another report demonstrated that TNFR2 expressed on nonhematopoietic cells is necessary for Treg function and suppression of EAE motor disease [127], suggesting that intrinsic and extrinsic TNFR2 activation impacts Treg functionality in EAE. Importantly, the absence of TNFR2 also resulted in the absence of solTNFR2 that can scavenge transmembrane TNF [128] and blocked its action, promoting a stronger TNFR1 signaling by solTNF. A recent study has addressed the role of TNFR2 in the context of the most common genetic variant linked to MS susceptibility, the HLA-DR2b (DRB1*15:01), by generating double transgenic mice expressing HLA-DR2b allele, thus selectively lacking MHC-II and TNFR2 in T cells and called DR2bΔR2 [129]. These transgenic mice developed a worse disease compared to their littermates lacking only MHC-II, due to increased development of Th17 cells. Data also showed that T cell-independent expression of TNFR2 plays a role in limiting astrocyte activation and neuroinflammation [129].

The individual roles of the two TNF-TNFR signaling pathways have also been investigated by pharmacological treatments in the EAE model, providing solid and convincing evidence that the specific inhibition of solTNF or TNFR1 is beneficial in EAE and may represent a valuable anti-TNF therapy for MS. The selective inhibition of solTNF through Xpro1595, which does not alter tmTNF signaling, has been studied by different groups and has proven a promising therapeutic strategy [130]. Treatment of EAE mice with Xpro1595 resulted protective in disease progression, while blocking both soluble and transmembrane TNF through etanercept failed to reduce clinical symptoms [119,120]. Along with these functional improvements, Xpro1595 produced a reduction in proinflammatory cytokine expression with preservation of structural fiber nerve composition [119] and promoted the expression of neuroprotective proteins [120]. Furthermore, it was demonstrated that the transmembrane signaling of TNF is essential to preserve the integrity and compaction of myelin and, above all, to promote remyelination in EAE mice [119,120].

Regarding TNFR targeting, the first compound used in EAE was an antagonist to TNFR1, PEGylated R1antTNF (PEG-R1antTNF) [131]. PEG-R1antTNF given in a preventive regimen suppressed EAE in a dose-dependent manner followed by a significant protection against on demyelination. In another study, a single injection of ATROSAB, a neutralizing antibody to TNFR1, at the time of immunization was sufficient to delay and significantly improve EAE symptoms and to promote neuroprotection in C57Bl/6 mice [132]. In addition, two sequential therapeutic injections of ATROSAB at disease onset significantly reduced EAE symptoms [132]. In another study, EAE was induced in humanized TNFR1 knock-in mice that were treated with ATROSAB antibody [132]. By this experimental approach the researchers observed beneficial effects in terms of disease severity, inflammatory cell infiltration into the CNS correlated with a significant reduction in demyelination and axonal damage. Of note, ATROSAB treatment was given every four days starting during preclinic phase of the disease indicating a crucial role of TNF in the onset of motor disability. Furthermore, long-term beneficial effects of TNFR1 inhibition were observed in a different cohort of TNFR1 knock-in mice by treating mice until 35 days after the onset of EAE [132]. Recently the generation of a nanobody-based selective inhibitor of human TNFR1 was tested in EAE showing that both prophylactic and therapeutic administrations of the compound ameliorate disease symptoms, reducing neuroinflammation in spinal cord and subsequent neurodegeneration [133].

In line with the idea of a beneficial role of TNFR2 signaling, exogenous activation of TNFR2 has been shown to ameliorate EAE disease. Treatment with solTNFR:Fc/p80 fusion protein reduced the clinical deficit of the first attack of relapsing-remitting EAE and decreased chemokine expression [134]. Moreover, activation of TNFR2 using antibody EHD2-sc-mTNFR2 in EAE mice delayed the day of onset of motor deficits and the development of motor disease and prevented weight loss after motor symptom onset [135]. These potential therapeutic effects of TNFR2 were associated to promotion of Treg activity and stability in the inflammatory environment: the resulting expansion of regulatory T cells reduced demyelination by promoting OPC proliferation [135].

All together, these results of pharmacological manipulation of TNF signaling mirror data obtained in conditional KO mice and clearly indicate divergent roles for the individual TNFRs in CNS autoimmunity; TNFR1 seems to promote CNS inflammation and demyelination and TNFR2 apparently acts limiting pathology, by eliminating autoreactive CD4+ T cells and macrophages, by maintaining Treg functionality, and by promoting remyelination.

Finally, TNF has been shown to promote the synaptic alterations typically affecting the brain of EAE mice and consisting in a potentiation of the glutamatergic transmission that has been proposed to contribute to neurodegeneration in EAE/MS brains [136]. TNF released by activated microglia derived from EAE mice increased the duration of glutamate postsynaptic events by promoting the phosphorylation, expression, and activity of glutamate AMPA receptors [136]. Of note, the administration of an AMPA receptor inhibitor ameliorated EAE course and reduced dendritic spine loss [136]. Accordingly, intracerebroventricular (icv) infusion of etanercept in EAE mice rescued the kinetic glutamatergic alterations observed in EAE striatal medium spiny neurons and ameliorated anxious behavior observed before disease onset [137]. The involvement of TNF in altered cognitive circuits in EAE mice has also been investigated by Habbas and coworkers (2015). Local increase of TNF in the hippocampal dentate gyrus of EAE mice activates astrocytic TNFR1, which in turn triggers an astrocyte-neuron signaling cascade that results in persistent functional modification of hippocampal excitatory synapses. Moreover, TNF failed to alter synaptic properties in TNFR1 KO mice, but re-expression of the receptor only in astrocytes restored the effect [138]. These findings highlight that TNF may contribute to the pathogenesis of EAE and MS, by acting not only on immunoregulatory and demyelinating processes, but also promoting synaptic damages and cognitive disturbances.

### 4.2. TNF Role in CPZ Model

CPZ model provides an ideal system to study neuroinflammation, demyelination and remyelination in the absence of autoimmune response [139,140,141]. Intoxication with CPZ, a copper chelating agent, results in a predictable course of OL death and microglial infiltration at the site of damage in the murine CNS, particularly the midline corpus callosum. The processes of de- and remyelination are further amplified or modified by inflammatory mechanisms, involving microglia and astrocytes [142]. Unlike EAE, T cell infiltration in this model is rare and the integrity of the BBB is maintained during CPZ treatment.

Like in EAE, in CPZ mice TNF was demonstrated to act in a bivalent way (Table 1). In their seminal paper, Arnett and colleagues pointed to a critical role of TNF in CNS demyelination processes [139]. TNF KO mice demyelinated similarly to WT mice but failed to remyelinate and the number of proliferating OPCs in lesions were reduced compared with WT mice. The failure of remyelination was attributed to TNFR2, since CPZ-treated mice deficient in TNFR2 but not in TNFR1 showed similar reduction in their capacity for remyelination respect to TNF KO [139]. This suggests that TNFR1 signaling mediates axon demyelination, while signaling via TNFR2 appears to be responsible for promoting oligodendrocyte proliferation and regeneration.

In line with these findings, a recent paper has demonstrated that local CNS production of solTNF inhibits the onset of remyelination and CNS repair in CPZ mice [143]. In particular, treatment with XPro1595, that selectively blocks solTNF, promoted the remyelination of demyelinated axons, resolved microgliosis, and halted progression of axon damage. The beneficial effects of solTNF blockade were associated with increased number of OPCs in lesions compared to both controls and etanercept-treated mice, and with an early remyelination as a result of improved clearance of myelin debris by phagocytes [143]. Furthermore, etanercept treatment had no effect on both initial demyelination and the subsequent recovery of myelin, indicating that TNF does not contribute to demyelination or remyelination in the CPZ model [143].

These results show that TNF plays a crucial role in pathological processes of CPZ model and, in agreement with data in EAE, reveal that the selective inhibition of solTNF prevents neuroinflammation and demyelination, by increasing the number of OPCs in the lesion sites and by promoting the clearance of myelin debris by phagocytic CNS macrophages.

### 4.3. TNF Role in the Theiler’s Murine Encephalomyelitis Virus (TMEV) Model

TMEV is a picornavirus which, when injected intracranially into susceptible mice, causes a chronic, inflammatory demyelinating disease. The histopathology of the virally-induced demyelination consists of mononuclear cell infiltration and myelin sheath damage limited to the white matter [144]. The initial CD4+ T cell-mediated immune response against chronic TMEV infection of the CNS causes significant damage to myelin, which in turn results in the activation of myelin-specific T cell clones [145]. Inflammatory processes induce the recruitment and activation of macrophages and TNF is among the products of activated monocyte macrophage cells. CD8+ T cells have been demonstrated to be important in viral clearance [146,147], and these cells may also be critical effector cells during the chronic TMEV-induced demyelinating phase of infection. The secretion of multifunctional cytokines such as TNF is likely to contribute to the further development of perivascular infiltration and CNS demyelination. In the TMEV model, it has been shown that the level of TNF producing cells correlates with the degree of demyelination during disease progression [148]. Specifically, increased numbers of TNF-producing cells were shown in the spinal cord of TMEV-infected mice and the administration of anti-TNF antibody suppressed TMEV disease [149]. The direct contribution of macrophages to autoimmunity was demonstrated by Katz-Levy and coworkers (2000). In particular, macrophages isolated from the spinal cords of TMEV-infected mice were able to present not only viral-specific antigens but also self-antigens to T cells, which may result in activation of autoreactive T cells and could initiate autoimmune disease [148].

These data suggest that TNF is most likely involved in the pathogenesis rather than protection against virally-induced inflammatory demyelinating disease and further studies should be addressed to evaluate the potential therapeutic role of specific TNFR antagonists in this animal model.

### 4.4. TNF in Multiple Sclerosis

#### 4.4.1. Evidence for TNF Involvement in MS Pathology in Post Mortem Studies

First evidence of the involvement of TNF in the neuroinflammatory process occurring in MS were provided by post mortem studies in MS patients, showing high cytokine levels in close proximity to active CNS lesions [12,88,150,151]. Moreover, transcriptional analysis of cortical gray matter (GM) tissues of PMS subjects showed the upregulation of both TNFR1 and TNFR2 [152]. Further immunocytochemical characterizations have shown that TNFR1 is mainly expressed by neurons and oligodendrocytes, while TNFR2 is mostly expressed by microglia and astrocytes in cortical GM [152]. However, the higher expression of genes linked to TNFR1-mediated necroptosis, a different form of necrotic cell death, suggested the prevalence of such TNF-mediated destructive pathway [152]. These results were in line with another study showing the activation of necroptosis in oligodendrocytes of MS brain lesions [153]. More recently, Veroni and colleagues (2020) found a significant upregulation of TNFR1 in subpial GM lesions and in both active and normal appearing white matter lesions. Interestingly, the authors found an even more increased expression of TNFR2 mRNA in white matter lesions, including chronic active lesions, without changes in the GM and in normal white matter areas surrounding the rim of active lesions, suggesting reparative mechanisms in the areas close to damaged zones [154].

#### 4.4.2. TNF Peripheral Levels and MS Diagnosis and Progression

Post mortem studies represent a worthwhile tool to deepen the pathophysiology of neurological disorders, but they are intrinsically devoid of prognostic value. Thus, different groups have turned their attention to detectable fluid TNF-signaling biomarkers in MS patients, showing differential expression of TNF depending on disease stage and providing evidence that TNF levels correlate with MS progression [155,156] (Table 2). With some exceptions [157,158,159], TNF has been shown to be increased in the blood and in mononuclear cells and T cell subsets of MS subjects [160,161,162,163,164,165]. In these studies, authors have reported elevations of TNF concentrations in CSF of MS patients at the time of clinical exacerbation indicating a direct link between TNF release and clinical parameters of disease progression. In particular, TNF levels were significantly increased in PMS subjects with high disease progression [165] and correlated negatively with cognitive deterioration in RRMS [166]. In another study, TNF serum levels were significantly increased in active RRMS compared to nonactive RRMS [167]. In addition, dendritic cells from peripheral blood samples of SPMS induced a unique IFN-ϒ and TNF-secreting Th1 phenotype [168]. Notably, a recent study has shown that TNF plasma levels are positively associated with MS diagnosis and, more interestingly, elevated solTNFR1 and low solTNFR2 have been proved good prognostic markers of disease progression in PMS [169]. While univocal results were not obtained in the case of RRMS subjects, it appears that peripheral TNF levels are increased in PPMS and correlate with disease progression and that the detection of solTNFRs could represent a useful prognostic marker.

#### 4.4.3. TNF Levels in the CSF of MS Patients

The detection of TNF in easily accessible fluids, like serum or plasma, is appealing in terms of clinical monitoring of patients, but it cannot entirely explain disease development. Recent technical progress in terms of test sensitivity has made possible the detection of the levels of cytokines and other molecules in the CSF, which is more informative of the brain activity than the blood (Table 2). Thus, in the last years the identification of a specific intrathecal inflammatory signature has greatly enhanced [170,171]. In line with the idea that the CSF better mirrors CNS neuropathology, a positive correlation between TNF liquoral level and neurological deterioration was shown in a cohort of PMS, while serum TNF levels of the same subjects did not correlate with disease progression [155]. Thus, the lack of correlation between individual CSF and serum concentrations of these cytokines suggest their possible synthesis in CNS. Moreover, in another cohort of patients the frequency of CD4+TNF+ cells in the CSF was significantly increased in MS subjects compared to other inflammatory disorders, while the same cells isolated in the peripheral blood of the same patients were not changed [172]. These data suggest a specific intrathecal T cell reactivation and cytokine production. In other cohorts, significantly higher levels of TNF were measured in the CSF of patients with SPMS compared to patients with stable MS [165,173]. Furthermore, CSF TNF levels correlated with degree of disease progression in patient with PMS MS, indicating a direct role of TNF in MS clinical disability [155,173].

Elegant studies of combined tissue characterization and CSF analysis, have provided compelling evidence that the released inflammatory/cytotoxic mediators into the CSF by immune cells in the subarachnoid space create an intracerebral milieu that sustains chronic compartmentalized inflammation and, at the same time, directly mediates and/or exacerbates cortical pathology and disease progression [174,175,176]. The increased expression of proinflammatory cytokines, including TNF, detected in meninges and CSF in cases of post-mortem MS, highly correlated with GM demyelination both at the time of diagnosis and death [175]. These results suggest that the inflammatory milieu generated by infiltrating immune cells in the subarachnoid space of the MS meninges, leads to changes in the balance of TNF signaling and to increased demyelinating and neurodegenerative pathology in the underlying GM.

Collectively these data indicate that TNF is more associated to PMS than RRMS and suggest that further investigations are needed.

#### 4.4.4. TNF and Excitotoxic Damage in MS

In MS, the detrimental effect of glutamate signaling in the CNS is widely recognized to contribute to neurodegeneration [8,177,178] As mentioned, several studies in human and experimental MS have shown that potentiation of the glutamatergic transmission is a pathological hallmark of MS and its animal model [8]. Excess glutamate is deleterious, since it induces excitotoxicity and massive loss of brain function. In human and rodent brain, glutamate homeostasis is normally maintained by a balance between this reaction and glutamate reuptake from the synaptic cleft by several transporters expressed on glial cells like as EAAT-1 and EAAT-2 expressed on astrocytes and oligodendrocytes [179,180]. Alteration of glutamate homeostasis contributes to axonal and oligodendroglial as well as synaptic pathology in MS [8,177]. Immunohistochemistry studies on post mortem tissue of MS patients revealed decreased expression of glutamate transporter EAAT2 [181,182] in oligodendrocytes in area adjacent to MS lesion. Moreover, CSF glutamate levels have been found increased in MS [183] and correlated with secondary disease progression [184].

Chimeric experiments have been carried out to address the causal link between TNF and excitotoxic damage in MS. TNF-enriched CSF from PMS subjects was incubated with corticostriatal slices to study the synaptic effect of inflammatory milieu on synaptic transmission [173]. Glutamatergic transmission recorded in striatal neurons was exacerbated by the CSF derived from PMS, while no effect was observed in the presence of CSF derived from control and RRMS patients. Furthermore, preincubation of the slices with etanercept, a nonselective TNF blocker, rescued the synaptic dysfunction induced by PMS CSF [173]. In a recent study we investigated synaptic alterations in MS using a heterologous chimeric model to address the synaptotoxic potential of T cells. T cells have been previously implicated in glutamatergic transmission potentiation in the brain of EAE mice [136] and thus likely implicated in excitotoxic damage. T lymphocytes isolated from the peripheral blood of RRMS patients exacerbated the glutamatergic transmission when incubated on mice brain slices. In particular, only lymphocytes from patients with acute inflammation, as evidenced by the presence of gadolinium-enhancing lesions at MRI, and expressing high levels of TNF, were able to induce synaptic alterations. Notably, coincubation with etanercept prevented these alterations, confirming that TNF was mainly responsible for these findings [185]. By these two chimeric approaches the glutamatergic excitotoxicity was shown to be mediated by TNF released in the CSF of PMS [173] and by blood-derived T cells of active-RRMS [185], suggesting that TNF of peripheral and local origin can trigger synaptotoxic events in RRMS and PMS, respectively.

Overall, findings in human MS suggest the involvement of TNF in MS pathology. Moreover, though not fully exploring the whole TNF signaling and its receptors, they indicate that TNF is more associated to PMS than RRMS and suggest that further investigations are needed.

## 5. Targeting TNF Signaling as a Therapy for MS

Based on the recognized role of TNF in autoimmune disorders, since 1988 different drugs targeting TNF signaling have been successfully introduced for the treatment of inflammatory disease such as rheumatoid arthritis, inflammatory bowel disease, Crohn’s disease and ankylosing spondylitis, psoriasis [186,187,188]. Over time, the pharmacological armamentarium has improved moving from the first pan-anti-TNF drugs, which resulted in a nonselective inhibition of TNF signaling, to the more recent TNFR1- and TNFR2-targeted macro- or nanomolecules [130].

Despite promising results in other autoimmune diseases, anti-TNF therapy has proven unsuccessful in MS. Following the encouraging results gained in rheumatoid arthritis [189] two female patients with rapidly progressive MS were treated intravenously with monoclonal anti-TNF antibody (cA2), now called infliximab [16]. Starting from the first infusion of cA2 in both patients the number of gadolinium-enhancing lesions increased and no improvement in disease severity was detected [16]. The treatment was ended soon. The same happened for the first MS clinical trial with another nonselective TNF inhibitor, lenercept, that led to an exacerbation of symptoms in the treated patients and the interruption of the study [190]. Lenercept is a recombinant TNF receptor p55 immunoglobulin fusion protein (solTNFR-IgG p55), which reduced clinical signs in a phase-II clinical trial in patients with rheumatoid arthritis [191]. A total of 168 patients, including clinically stable RR and SPMS were enrolled in the study. While no significant differences were observed in terms of degree of expanded disability status scale (EDSS) and MRI data, lenercept-treated patients experienced significant exacerbations, meaning the appearance of new symptoms, and neurological deficits tended to be more severe compared to placebo-group, indicating that TNF inhibition was deleterious in MS. Of note, these effects were dose-dependent, being increased over the placebo by 2%, 68%, and 50% at lenercept doses of 10, 50, and 100 mg, respectively. Thus, increased frequency and severity of MS attacks in treated patients led the trial to the conclusion that lenercept was contraindicated as a therapy in MS [190].

Based on current knowledge of TNF biology, a possible explanation for these results, which at that time were quite unexpected, could be linked to the pleiotropic actions of TNF, including both pro- and anti-inflammatory functions as well as the almost ubiquitous expression and action of such cytokine. Moreover, the relationship between nonselective TNF inhibitors and MS is further supported by several cases of MS onset observed in patients with other diseases [192,193]. Patients with rheumatoid arthritis treated with either etanercept or infliximab showed new-onset neurologic signs and symptoms, associated with demyelinating lesions of the CNS [194] or symptoms fulfilling the criteria for clinically definite RRMS [195]. More recently, a case report showed a 51-year old male patient treated for a psoriasis arthritis with an anti-TNF agent, adalimumab, developed slowly progressing neurological deficits about 18 months after treatment initiation [192]. Overall these studies suggested that nonselective inhibition of TNF is harmful both in MS and in other inflammatory disease, but at the same time highlight the relevance of TNF in the pathophysiology of MS. Current guidelines suggest to avoid the use of anti-TNF drugs in individuals with a history of MS or demyelinating disease and patients with inflammatory disease should be carefully monitored with frequent MRI regardless of their clinical status in order to prevent the occurrence of MS.

In order to clarify the effect of anti-TNF therapy on MS patients and non-MS autoimmune diseases, genome-wide association studies (GWAS) may help in identifying a link between the specific response to anti-TNF therapy and the presence of the single nucleotide polymorphism in TNF-related genes that may affect TNF signaling. In particular, two polymorphisms of the TNFR1-encoding gene, TNFRSF1A, the rs1800693 with high frequency and the rs4149584 with low frequency, have been identified and associated with an increased risk for MS and not for other autoimmune disorders [196]. The research has focused on the rs1800693 variant, showing that this polymorphism promotes the expression of solTNFR1, which acts as a de facto natural TNF-scavenger [197]. Further studies have confirmed that both rs1800693 and rs4149584 polymorphisms have consequences in TNFR1 function in peripheral blood cells, like monocytes and T cells [198,199]. Besides, two additional genetic polymorphisms linked to TNF signaling and associated to MS, deserve attention. It has been shown that both a genetic variant proximal to NFkB1 gene (rs228614-G) and one in an intron of TNFRSF1A gene (rs1800693-C) are associated with increased NFkB signaling after TNF stimulation [200]. These data strengthen the idea that genetic screening would be of great help in determining which patients (MS or not) are suitable for a therapy targeting TNF signaling.

## 6. Conclusions and Perspectives

As reviewed, TNF is an important mediator of disease in MS playing a key role in the pathogenesis of the main hallmarks characterizing MS and its related preclinical models. Although clinical trial with nonselective anti-TNF antibodies completely failed and discouraged neurologists from using anti-TNF agents for MS patients, in the light of the recent advances in the knowledge of TNF signaling, the anti-TNF therapy for MS treatment should be re-examined. On one hand, data from MS models in TNF KO mice or in mice treated with pan-anti-TNF drugs, agree with human studies. On the other hand, recent studies in experimental MS provide clear evidence of the complexity of the TNF/TNFRs signaling in MS, clarifying some crucial aspects of the disease and opening the possibility to test novel and more selective compounds. In particular, the opposite immunoregulatory effects of TNFRs suggest that the low efficacy of nonselective TNF blockers (like etanercept) in the EAE model is probably due to the abrogation of TNFR2 neuroprotective action and leads to the speculation that selective regulation of TNFR1 and TNFR2 may be a new molecular target for the development of therapeutic agents in MS. Testing anti-TNF signaling drugs in animal models of MS in combination with genetic ablation of TNFRs in specific cellular population, could be of great value to distinguish the role of each TNFRs in a specific cellular context. Indeed, the use of selective conditional KO mice has highlighted subtle but relevant differences in the function of TNFR1 and TNFR2 in a cell- and context-dependent manner [116,126]. Moreover, as perceptively reviewed elsewhere [130,201], new molecules designed to target TNF signaling have been developed and are under investigation in several disease models, including neurodegenerative diseases. It is worth noting that the selective solTNF inhibitor XPro1595, one of the best studied compounds in experimental MS, has recently entered a phase-1 clinical trial in mild to moderate AD patients (ClinicalTrials.gov identifier-NCT number: NCT03943264).

Based on the literature reviewed here, a dichotomic, albeit simplistic, vision of TNF signaling and targeting in MS can be proposed. Considering that TNFR1 exerts inflammatory and proapoptotic functions, whereas TNFR2 has a neuroprotective function, the combination of inhibitors of TNFR1 signaling and TNFR2 agonists represents promising a therapeutic approach for the treatment of MS (Figure 1). The promotion of cell survival and proliferation mediated by TNFR2 may be able to treat acute neurodegenerative disorder caused by glutamate-induced excitotoxicity and to promote remyelination. Furthermore, TNFR2 agonists are particularly appealing, in the light of the remarkable involvement of this receptor in Treg activity and of the breakdown of Treg function in MS. To this respect, we envisage the need to test these new molecules (alone or in combination with TNFR1 inhibitors) in animal models of MS, enhancing the research also in the less studied Cuprizone and TMEV models.

## Figures and Tables

**Figure 1 cells-09-02290-f001:**
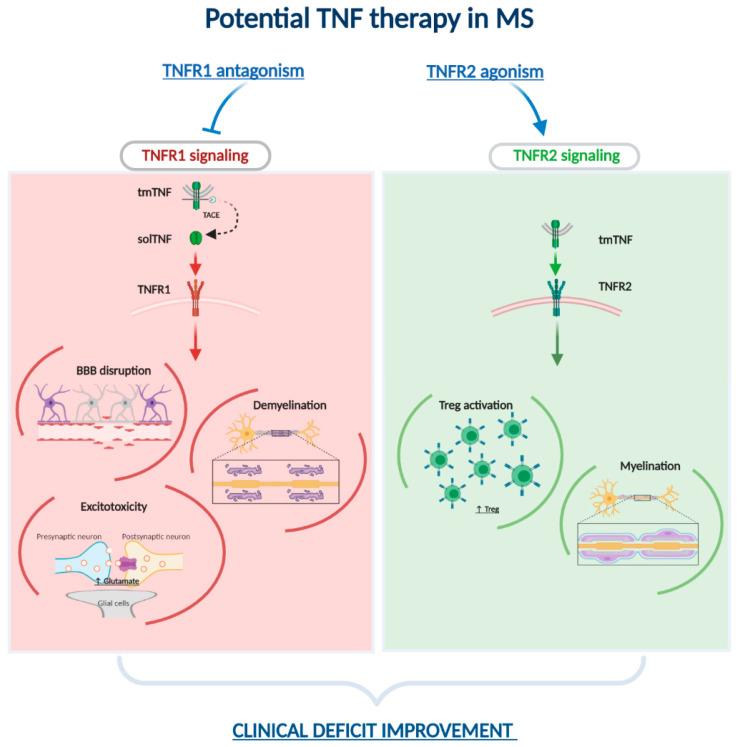
Potential working model of TNF therapy in MS. TNF exists in two different forms, tmTNF (transmembrane) and solTNF (soluble), this latter synthetized by TACE enzymatic activity and thus released only by TACE expressing cells. solTNF binds preferentially TNF receptor type-1 (TNFR1) while tmTNF recognizes type-2 (TNFR2). TNF and its receptors are involved in many pathological processes of MS. Signaling through TNFR1 is likely responsible for BBB disruption, demyelination and glutamate-excitotoxity, while TNFR2 activation by tmTNF is supposed to promote myelination and Treg expansion and activation. The combination of molecules selectively modulating TNFRs signaling through TNFR1 antagonism and TNFR2 agonism is proposed as a putative novel approach for MS treatment. BBB, blood–brain barrier; Tregs, regulatory T cells; TNF, tumor necrosis factor; TACE, TNF converting enzyme.

**Table 1 cells-09-02290-t001:** Genetic and pharmacological modulation of tumor necrosis (TNF) signaling in animal models of multiple sclerosis (MS).

EAE Model					
	Clinical Score		BBB Permeability	Demyelination	Synaptic Dysfunction
	Onset	Severity			
TNF-KO	↑ 111; 112; 113; 116; 123; 122; 115 = 115 (T cell) 129	↑ 115; 114; 122 = 111; 112; 113 ↓ 129; 123; 116 115 (T cell)	↓ 116	↑ 114; 122 = 111; 112 ↓ 123 115 (T cell)	
anti-TNF therapy	= 120 (xpro1595- etanercept)	= 119; 120 (xpro1595- etanercept)		= 119 (xpro1595- etanercept) ↓ 120 (xpro1595- etanercept)	↓ 137 (etanercept) 120 (xpro1595-etanercept) 136 (TNFR-Ig-ex vivo) 138
TNFR1-KO	↑ 122; 123; 133	↓ 122; 123; 133		↓ 122; 123; 133	↓ 133
TNFR1-antagonism	↑ 133 (Nanobody) = 120 (xpro1595- etanercept) 132 (ATROSAB)	↓ 119; 120 (xpro1595-etanercept) 132 (ATROSAB) 133 (Nanobody) 117 (TNFRp55-IgG)		↓ 119 (xpro1595-etanercept) 132 (ATROSAB) 133 (Nanobody)	↓ 119 (xpro1595-etanercept) 132 (ATROSAB) 133 (Nanobody)
TNFR2-KO	↑ 126 (monocytes) = 122; 98; 125; 129 ↓ 126 (microglial)	↑ 129 122; 98; 123; 125; 126 (microglia) ↓ 126 (monocytes)		↑ 98; 123; 122; 125; 126 (microglia) ↓ 126 (monocytes)	↑ 98; 125;126 (microglia) ↓ 126 (monocytes)
TNFR2-agonism	↑ 134 (sTNFR:Fc/p80)	↓ 134 (sTNFR:Fc/p80) 135 (EHD2-sc-mTNFR2)		↓ 135 (EHD2-sc-mTNFR2)	↓ 135 (EHD2-sc-mTNFR2)
**Cuprizone Model**					
		**Demyelination**		**Remyelination**	
TNF-KO		= 139		↓ 139	
anti-TNF therapy		= 143		= 143	
TNFR1-KO		= 139		↑ 139	
TNFR1-antagonism		= 143		↑ 143	
TNFR2-KO		= 139		↓ 139	

**↑** increase; **↓** decrease; =no effect. Abbreviations: TNF, tumor necrosis factor; TNFR, TNF, tumor necrosis factor receptor; EAE, experimental autoimmune encephalomyelitis; BBB, blood–brain barrier; CPZ, cuprizone.

**Table 2 cells-09-02290-t002:** TNF levels in MS patients.

TNF Levels	Serum	CSF	Brain
Active-RRMS	↑ 160; 161; 162; 163; 164; 165 166; 167		
PPMS	↑ 160; 161; 162; 163; 164; 165	↑ 155; 165; 170; 171; 173	↑ 174; 175; 176
SPMS		↑ 164; 173	

**↑** Increase. Abbreviations: TNF, tumor necrosis factor; RRMS, relapsing–remitting multiple sclerosis; PPMS, primary progressive multiple sclerosis; SPMS, secondary progressive multiple sclerosis; CSF, cerebrospinal fluid.

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
