# Peer review of "Re-Examining the Role of TNF in MS Pathogenesis and Therapy"

_cells, 2020, doi:10.3390/cells9102290_

Round 1

Reviewer 1 Report

Review cells

This is an interesting review on a difficult and contentious topic. While carefully assembled and referencing largely important and recent work I have various issues with this manuscript

Large parts (especially chapter 1 and 2) are not addressing the topic and should be shortened.

EAE should not be addressed as "preclinical" model of MS. It is an animal model of the early inflammatory stages of MS. Many treatments work in EAE but fail in MS. The limitations should be discussed in more detail.

It has been shown repeatedly (quoted by the authors) that anti-TNF therapy does work in EAE but not in MS. In contrast it might even exacerbate the disease. One wonders why this review tries to revive the idea of such a treatment. This needs to be explained better at the end of the review.

Author Response

Reviewer 1 Comments:

This is an interesting review on a difficult and contentious topic. While carefully assembled and referencing largely important and recent work I have various issues with this manuscript.

  1. We thank the reviewer for the positive comments and for the issues raised below.

Large parts (especially chapter 1 and 2) are not addressing the topic and should be shortened.

  1. We agree with the reviewer and we have made some edits and cut to make this part as much as possible simple and introductory to the main theme. Changes are highlighted in red in the text. See page 2-3, lines 53-59,64-68,78-81 and 101-106 of the new version of the manuscript. Moreover, in compliance with reviewer 2 suggestion, paragraph 3.2.1 and 3.2.3 have been deleted and only few concepts related to the topics discussed in those paragraphs have been included in paragraph 3.2. See page 4-5, lines 177-181 and 213-226 of the new version of the manuscript.

EAE should not be addressed as "preclinical" model of MS. It is an animal model of the early inflammatory stages of MS. Many treatments work in EAE but fail in MS. The limitations should be discussed in more detail.

  1. We understand the criticism raised by the reviewer about the reliability of the EAE model. Several discussions have been made around the validity of this model. However, with all the intrinsic limitations of this animal model, it is widely accepted that research in EAE has been instrumental in the discovery of many immunomodulatory drugs used for MS, such natalizumab, glatiramer acetate. To comply with the reviewer criticism, we have changed the text at page 5, lines 244-247 of the new version of the manuscript and added a reference on this (ref 100).

It has been shown repeatedly (quoted by the authors) that anti-TNF therapy does work in EAE but not in MS. In contrast it might even exacerbate the disease. One wonders why this review tries to revive the idea of such a treatment. This needs to be explained better at the end of the review.

  1. The ultimate goal of this review is to propose a re-examination of the anti-TNF therapy in MS. In MS only pan-TNF inhibitors have been tested, while recent studies in animal models of MS have been focused on more selective inhibitors, showing protective and beneficial effects against (see references for the solTNF inhibitor XPro1595 or TNFR1 antagonists). In some papers, non-selective inhibitors of TNF, like etanercept, have been shown to worsen disease symptoms in EAE, in agreement with human studies. Based on these, we have added some additional sentences in the conclusion to highlight the need to better characterize the effects of selective TNF inhibitors in different animal model of MS. See page 16-17, lines 628-632 and 636-645 of the new version of the manuscript.

Reviewer 2 Report

The review comprehensively summarizes recent findings on TNF and its role in multiple sclerosis, providing an updated description of current knowledge in the field. The text is easy to follow and provides a critical discussion of reported data. The introductory part provides the essential information to frame the topic for readers non in the field. I really enjoyed reading some paragraphs, such those on the role of TNF on immune system and Treg cells, on genome-wide association studies and the response to anti-TNF therapy. However, too many topics are probably addressed by the review, making it too long and a bit boring. Despite related to the role of TNF in MS, my suggestion is to eliminate the paragraph 3.1 and 3.2 on BBB and neuroinflammation and myelination, which are not really well developed, in order to keep the review more concise. Other manuscripts in the TNF special issue will probably focus on these subjects

The two tables properly summarize all the studies on TNF carried out so far in the EAE and cuprizone models of MS as well in human body fluids of patients. The figure nicely depicts the mechanisms which may mediate the therapeutic action of a novel and more complex approach targeting TNF

Below a few minor suggestions and typos:

line 50: includeS

 Line  150  please check ref 44. It seems the wrong one

line 126: activateS

Line 224 please define the time of prolonged activity blockade to compare this finding with that described in the next sentence

line 295: the meaning of the sentence is not clear.

Frase 371: missing a word after anti-TNFr1. Inhibitors?

Line 549 please introduce etarnecept to the readers

Line  454: "the administration of anti-TNF antibody suppressed TMEV...? infection? function?

Line 500: remove "not"

Lines 563-4 based on data described above, it is not clear to me why TNF is more associated to PMS

Lines 593-596 Please check the sentence for clarity. It is not clear to me why the onset on MS in patients treated with pan TNF inhibitors is in agreement with demyelination in mice overexpressing the cytokine

line371: missing a word after anti-TNFr1. Inhibitors?

line 500: remove "not"

Author Response

Reviewer 2 Comments:

The review comprehensively summarizes recent findings on TNF and its role in multiple sclerosis, providing an updated description of current knowledge in the field. The text is easy to follow and provides a critical discussion of reported data. The introductory part provides the essential information to frame the topic for readers non in the field. I really enjoyed reading some paragraphs, such those on the role of TNF on immune system and Treg cells, on genome-wide association studies and the response to anti-TNF therapy.

  1. We thank the reviewer for the positive comments.

However, too many topics are probably addressed by the review, making it too long and a bit boring. Despite related to the role of TNF in MS, my suggestion is to eliminate the paragraph 3.1 and 3.2 on BBB and neuroinflammation and myelination, which are not really well developed, in order to keep the review more concise. Other manuscripts in the TNF special issue will probably focus on these subjects

  1. We agree with the reviewer and we have cut the paragraphs on BBB and neuroinflammation and myelination. Now, paragraph 3 is divided into two main sub-paragraphs, i.e. 3.1 TNF role in immune system and 3.2 TNF role in the brain. However, for clarity we have included some information regarding these topics at the end of paragraph 3.2. See page 4-5, lines 177-181 and 213-226 of the new version of the manuscript.

The two tables properly summarize all the studies on TNF carried out so far in the EAE and cuprizone models of MS as well in human body fluids of patients. The figure nicely depicts the mechanisms which may mediate the therapeutic action of a novel and more complex approach targeting TNF

  1. We thank the reviewer for the positive comments. In compliance with the Academic Editors we have included some additional studies addressing the role of TNF and TNF therapy in EAE model. See page 9, lines 397 of the new version of the manuscript.

Below a few minor suggestions and typos:

line 50: includeS

R: Changed. See page 2, lines 50 of the new version of the manuscript

Line 150  please check ref 44. It seems the wrong one

R: Changed. See page 5, lines 223 check ref. 97 of the new version of the manuscript

line 126: activateS

R: Changed. See page 3, lines 105 of the new version of the manuscript

Line 224 please define the time of prolonged activity blockade to compare this finding with that described in the next sentence

R: we added this information (48 h) in the text. See page 5, line 194 of the new version of the manuscript.

line 295: the meaning of the sentence is not clear.

R: We have changed the text. See page 6, line 250 of the new version of the manuscript.

Frase 371: missing a word after anti-TNFr1. Inhibitors?

R: We have changed with “ATROSAB”. See page 8, line 356 of the new version of the manuscript.

 Line 549 please introduce etarnecept to the readers

R: we introduced etarnercept only as a “non-selective blocker “ as it is specified on line 291-292 of the new version of the manuscript. See page 15, line 551 of the new version of the manuscript.

Line 454: "the administration of anti-TNF antibody suppressed TMEV...? infection? function?

R: the administration of anti-TNF antibody suppressed TMEV disease. See page12, line 448 of the new version of the manuscript.

Line 500: remove "not"

R: Changed. See page 13, lines 495 of the new version of the manuscript

Lines 563-4 based on data described above, it is not clear to me why TNF is more associated to PMS

R: We thank the reviewer for this comment. Indeed, the above sentence was in the wrong position, being more related to the conclusion of the concept described in paragraph 4.4.3. Therefore, the sentence has been moved to the end of such paragraph to facilitate the comprehension of the text. See page 14, line 522-523 of the new version of the manuscript.  

Lines 593-596 Please check the sentence for clarity. It is not clear to me why the onset on MS in patients treated with pan TNF inhibitors is in agreement with demyelination in mice overexpressing the cytokine

R: We thank the reviewer for highlighting this misleading sentence. We have deleted the second part of the sentence. See page 16, line 595-596 of the new version of the manuscript.